# Body Composition and Senescence: Impact of Polyphenols on Aging-Associated Events

**DOI:** 10.3390/nu16213621

**Published:** 2024-10-25

**Authors:** Tanila Wood dos Santos, Quélita Cristina Pereira, Isabela Monique Fortunato, Fabrício de Sousa Oliveira, Marisa Claudia Alvarez, Marcelo Lima Ribeiro

**Affiliations:** 1Laboratory of Immunopharmacology and Molecular Biology, Sao Francisco University, Av. Sao Francisco de Assis, 218, Braganca Paulista 12916-900, SP, Brazil; tanilawood@gmail.com (T.W.d.S.); quelitapereirapa@gmail.com (Q.C.P.); fortunato.misabela@gmail.com (I.M.F.); fabricio1445@hotmail.com (F.d.S.O.); marisacalvarez@yahoo.com (M.C.A.); 2Hematology and Transfusion Medicine Center, University of Campinas/Hemocentro, UNICAMP, Rua Carlos Chagas 480, Campinas 13083-878, SP, Brazil

**Keywords:** body composition, adipose tissue, skeletal muscle, senescence, polyphenols

## Abstract

Aging is a dynamic and progressive process characterized by the gradual accumulation of cellular damage. The continuous functional decline in the intrinsic capacity of living organisms to precisely regulate homeostasis leads to an increased susceptibility and vulnerability to diseases. Among the factors contributing to these changes, body composition—comprised of fat mass and lean mass deposits—plays a crucial role in the trajectory of a disability. Particularly, visceral and intermuscular fat deposits increase with aging and are associated with adverse health outcomes, having been linked to the pathogenesis of sarcopenia. Adipose tissue is involved in the secretion of bioactive factors that can ultimately mediate inter-organ pathology, including skeletal muscle pathology, through the induction of a pro-inflammatory profile such as a SASP, cellular senescence, and immunosenescence, among other events. Extensive research has shown that natural compounds have the ability to modulate the mechanisms associated with cellular senescence, in addition to exhibiting anti-inflammatory, antioxidant, and immunomodulatory potential, making them interesting strategies for promoting healthy aging. In this review, we will discuss how factors such as cellular senescence and the presence of a pro-inflammatory phenotype can negatively impact body composition and lead to the development of age-related diseases, as well as how the use of polyphenols can be a functional measure for restoring balance, maintaining tissue quality and composition, and promoting health.

## 1. Introduction

Human longevity is closely associated with a progressive decline in the repair potential of organs and tissues, as well as their regenerative capacity. Environmental, genetic and epigenetic factors are involved in decreasing the conditions that combat physiological stress at the molecular, cellular, and systemic levels, through the complex and interactive molecular mechanisms that are activated to promote aging [1].

This process is complex, driven by several changes known as the hallmarks of aging, among which cellular senescence plays a crucial role [2,3,4]. Cellular senescence has been directly implicated as a key factor in aging, with significant effects on the maintenance of normal tissue homeostasis, as well as in pathological conditions, being considered the main factor driving the aging process and increasing susceptibility to chronic diseases [1].

Senescent cells are characterized by cell cycle instability and a loss of proliferative capacity, even in the presence of mitogenic stimuli [5]. These findings began in 1961 with a continuous culture of human diploid cells, where it was first discovered that the lifespan of fibroblasts was limited. The study revealed that fibroblasts were unable to divide after 40–60 population doublings [6]. Beyond the replicative senescence caused by the shortening of telomeres and the chronic stimulus in response to DNA damage, cellular senescence can arise from other stressful conditions, such as epigenetic changes, genomic instability, mitochondrial dysfunction, oxidative stress, reactive metabolites, inactivation of certain tumor suppression genes, oncogenic stress, and viral infections [6,7,8,9,10,11].

Body composition tissues, which can be understood as muscle and adipose tissues, are profoundly affected by such changes, and the resulting metabolic and structural modifications have been strongly associated with the development of diseases in elderly individuals [12,13,14].

In light of this, the use of therapeutic strategies that can mediate events such as cellular senescence may be a promising preventive measure for promoting healthy aging [1,15,16,17]. The use of bioactive natural substances with therapeutic potential has been extensively studied, and their anti-senescence properties are increasingly being explored in the current literature [18,19,20,21,22,23,24,25,26].

This review aims to address the impacts of senescence on body composition and the possible metabolic changes resulting from this process, as well as how polyphenols can act beneficially in this context.

## 2. Cellular Senescence

In recent decades, studies carried out by Lopez-Otín et al., identified the typical characteristics of aging that gave rise to the current “hallmarks of aging”, among which cellular senescence plays an important role and can serve as a link between the other well-known hallmarks of aging [3,4].

Cellular senescence is characterized by a profoundly altered cellular state at a morphological, biochemical, and metabolic level, in which cells permanently lose the capacity for cell division and enter a state of stable cell cycle arrest. Furthermore, at a molecular level, these cells present shortened telomeres, an altered chromatin structure, an accumulation of DNA damage, an increase in reactive oxygen species (ROS), the activation of cell cycle inhibitory pathways, such as p53, p16^Ink4a^ and/or p21^CIP1^, senescence-associated β-galactosidase (*SA-β-gal*) activity, and a resistance to apoptosis as well as the development of senescence-associated heterochromatic foci [27,28].

Furthermore, the factors that induce the cellular senescence process, such as genotoxic agents, inflammatory factors, and metabolic stress, also contribute to the development of the senescence-associated secretory phenotype (SASP). This phenotype is responsible for mediating the pro-inflammatory immune responses that induce the propagation of cellular senescence. Although cellular senescence is a natural mechanism and is also associated with positive effects, such as simulating wound healing after tissue damage, and contributing to tumor suppression, its accumulation in tissues and at a systemic level result in organic dysfunctions and aging. In fact, the cumulative cellular damage that occurs with age, at a given moment, can affect proliferating cells, culminating in the senescence of this cell type, coinciding with a reduction in the clearance of senescent cells by macrophages [29,30] (Figure 1).

## 3. Changes in Body Composition Accompanying Aging

The aging process is involved in changes in body composition, particularly in fat mass and lean mass. These changes can result in worsening health conditions, especially in middle and old age. In general, in human individuals the proportion of fat mass tends to increase with advancing age, while a simultaneous reduction in fat-free mass is observed [31].

A study conducted by Health ABC created a paradigm shift in the way the aging process was previously understood [32]. This study highlighted the magnitude of the impact of body composition on the dynamics of health conditions as well as the physiological functions of elderly individuals, highlighting that both fat and muscles are important. It also emphasized that the aspects inherent to muscle quality, such as strength per mass and fat infiltration, play a critical role in the pathophysiology of sarcopenia [14].

The dysfunctions associated with skeletal muscle are among the main causes of morbidity in the elderly worldwide [33,34]. The damage caused by sarcopenia, a loss of muscle mass and strength, is associated with an increased risk of disability and mortality in elderly individuals, as well as significant healthcare costs [34,35,36]. Sarcopenia does not yet have a consensual definition criterion; however, most of the criteria used include measurements of grip strength, walking speed, and lean mass. Currently, according to the International Classification of Diseases, sarcopenia is classified as a pathology [34,37].

In addition to the decline in lean mass and bone density, another important change in body composition observed with advancing age refers to the increase in the proportion of fat mass, particularly in the abdominal area [34,38,39]. Increased fat mass is associated with an increased risk of physical disability and all-cause mortality in the elderly, although some protective effects of moderate excess weight (BMI 25 ≤ BMI < 30) have been reported in relation to mortality in meta-analyses [39,40,41,42,43,44]. Paradoxically, large studies have pointed to obesity (BMI ≥ 30) and central adiposity (indicated by a higher waist-to-hip ratio) as risk factors for coronary heart disease, type 2 diabetes, and mortality [45,46,47]. Furthermore, the changes in body composition that occur with age, represented by a decline in lean mass and a concomitant increase in fat mass, can result in the establishment of sarcopenic obesity, causing more harm to health than sarcopenia or obesity alone [48].

Data from cohort studies have pointed to the inflammatory process as an important factor associated with the development of various imbalances, such as physical function deterioration, periodontal disease, and depression [14,32,49,50,51]. These dysfunctions, whether associated or individual, have been indicative of more severe prognoses, such as cardiovascular disease, heart failure, incident pneumonia, lung cancer, cognitive decline, and incident mobility limitation [14,52,53,54,55,56]. Furthermore, the number of elevated inflammatory markers has shown associations with multiple diseases, and is also strongly associated with the loss of muscle mass and strength [57,58,59]. These associations collectively support the role of inflammation as one of the important hallmarks of aging [3,60].

Moreover, the systemic chronic inflammation associated with aging is also accompanied by cellular senescence and immunosenescence, which strongly contribute to the establishment of the aforementioned organ dysfunctions and diseases [61]. The tissue response of each body compartment involves different metabolic responses that culminate in the development of the chronic conditions associated with aging, among which are diabetes and cardiovascular diseases, as well as the limiting conditions such as sarcopenia, which have a profound impact on the overall health conditions of the elderly population.

In summary, as mentioned above, it seems evident that aging causes increased fat mass and decreased lean mass, leading to health issues such as sarcopenia, obesity, and an increased risk of chronic diseases like diabetes and cardiovascular conditions. The decline in muscle strength and the rise in abdominal fat are associated with greater disability and mortality. Inflammation and cellular senescence are key factors in these changes, underscoring the need for effective interventions to address the multifaceted impacts of aging on health.

## 4. Body Composition and Senescence

Cellular senescence is known as one of the main causes of age-related dysfunctions, and is marked by morphological changes, such as enlarged nucleoli as well as increased positivity for senescence-associated beta-galactosidase (SA-β-gal) [62,63,64].

The characterization of the cellular senescence process includes a variety of cellular stresses that can result in cell cycle arrest in proliferating and long-lived cells, and the consequent associated phenotypic and functional changes [65,66,67]. Many of these changes are dependent on constitutive signaling through the mTOR, which promotes protein synthesis and increased skeletal muscle mass, leading to hypertrophy [68,69,70]. Furthermore, senescent cells are also resistant to apoptosis, and exhibit local DNA methylation changes and global chromatin rearrangements, leading to altered patterns of gene expression and the secretion of a variety of cytokines, chemokines, and tissue remodeling enzymes [71,72,73].

The process of cellular senescence involves changes in the p21/p53 and p16 pathways and, although the p53/p21 is essential for initiating cell cycle arrest, p16 is recognized as important for maintaining the arrest and thus reinforcing the maintenance of senescence [74]. Furthermore, the production of a senescence-associated secretory phenotype (SASP) by senescent cells results in the chronic release of pro-inflammatory cytokines and chemokines which promote the recruitment of leukocytes for tissue repair and remodeling, as well as increased generation of reactive oxygen species (ROS) [64,74,75,76,77].

### 4.1. Mechanisms of Adipose Tissue Dysfunction During Aging

Adipose tissue is one of the most vulnerable tissues in aging, affected by a variety of changes in biological and physiological processes, which compromise the body’s general wellbeing. In this sense, a study showed for the first time that an age-related immune response was detected in white adipose depots [78]. Since adipose tissue is a source of stem cells of great importance for regenerative transplantation, treatments that are able to reverse age-related dysfunctions in adipose tissue-derived stem cells could improve the efficiency of stem cell therapy [12,79]. As the largest endocrine and energy storage organ, adipose tissue plays a significant role in metabolism and energy homeostasis. Thus, the dysfunctional adipose tissue in aging promotes chronic low-grade inflammation, insulin resistance, and lipid infiltration in the elderly [80,81,82].

Metabolic dysfunction in adipose tissue is probably caused by cellular senescence in adipose tissue, as shown by the findings that show the inhibition of p53 activity in adipose tissue significantly improved insulin resistance [83,84]. In addition, it has been shown that the inhibition of senescent cells or their products in adipose tissue improves the metabolism of aged mice [85]. More evidence indicates that the changes in adipose tissue that occur with aging contribute to the development of insulin resistance in the elderly. Dysfunctions in the insulin signaling cascade caused by advancing age, such as reduced insulin-stimulated tyrosine phosphorylation, are more severe in adipose tissue than in liver or muscle tissue, suggesting that adipose tissue may be a source of insulin resistance during aging [12,86,87,88,89]. Furthermore, the adipose redistribution and chronic inflammation derived from aging adipose tissue result in metabolic disorders, including insulin resistance, impaired glucose tolerance, and diabetes. The increased production of pro-inflammatory cytokines in dysfunctional adipose tissue, such as members of the IL-1 family, is directly associated with the disruption of the insulin signaling pathway [90,91]. Complementarily, the changes in immune system cells that occur during aging, such as the accumulation of T cells, are also associated with the development of insulin resistance [92,93,94].

The accumulation of senescent cells can be induced by various endogenous and exogenous stresses, including DNA damage, telomere shortening, oncogenic mutations, and environmental stresses [95]. It has been established that cellular senescence is an important mechanism for preventing cancer progression, and that the p53 and pRB tumor pathways are central regulators of senescent cell accumulation [96]. In this sense, it has been demonstrated that p53 inhibition induces senescent cells to re-enter the cell cycle [12,97,98]. Furthermore, chronic inflammation with a continuous upregulation of pro-inflammatory mediators is also associated with the development of cellular senescence, which in turn becomes a source of pro-inflammatory secretion [75,99,100,101]. SASP factors derived from senescent cells in adipose tissue contribute to the exacerbation of the inflammatory process through the overstimulation of the production of the following: cytokines, chemokines, and TNF receptors; growth factors such as EGF, VEGF, and NGF and the extracellular matrix (ECM); adipose macromolecules such as fibronectin, in addition to collagen and laminin, inducing tissue remodeling; and soluble non-protein factors, such as nitric oxide, in the microenvironment [102,103].

The central signaling pathway for SASP generation is also likely shared among the different types of senescent cells, converging on the transcription factor NF-κΒ, an important regulator of inflammation in immune cells that also plays a critical role in the initiation of a SASP [104,105,106]. In this sense, the activation of the NFκB signaling pathway induces the polarization of M1 macrophages, which is pro-inflammatory in adipose tissue. It is important to highlight that this polarization occurs independently of the total tissue mass [107]. Additionally, endoplasmic reticulum stress may also be a causal factor in stimulating M1 macrophage polarization, as it promotes a pro-inflammatory environment due to the secretion of cytokines such as IL-6, MCP-1, and TNF-α in this tissue [108].

Studies have demonstrated that the endocrine profile of visceral adipose tissue, as well as other deposits, such as perivascular, interscapular, and perirenal, becomes more pro-inflammatory with aging, as evidenced by the increased expression of cytokines TNF-α and IL-6 in murine models of aging when compared with young animals [109,110]. In humans, the monocyte production of cytokines, such as IL-6 and IL-1Ra, but not IL-1β or TNF-α, was increased in the elderly compared to the young healthy individuals [111]. Complementarily, the presence of pro-inflammatory cytokines, such as TNF-α, IL-1β, IL-6, and MCP-1 secreted by adipose tissue macrophages, was also associated with the presence of senescent cells (p16INK4A), resulting in immunosenescence [107,112,113,114] (Figure 2).

### 4.2. Mechanisms of Skeletal Muscle Tissue Dysfunction During Aging

Skeletal muscle is capable of adapting to different stress conditions, including pathological ones, through metabolic changes, such as modifications in the size and composition of fibers [13]. The presence or absence of these adaptations during aging are decisive in the homeostatic maintenance of this tissue and can result in atrophy or significant loss of muscle mass, and consequent health problems for the elderly [57].

The dysfunction associated with muscle loss represents one of the most problematic changes that occur with aging, leading to dramatic effects on an individual’s autonomy and quality of life [115]. Age-related muscle loss has been studied since the early 1970s [116]. Since then, prospective studies have sought more precise estimates of the decline in annual skeletal muscle mass, finding an estimated proportion between 0.65% and 1.39% for elderly men and between 0.61% and 0.80% for elderly women, using dual-energy radiological absorptiometry (DXA) [117].

Satellite cells (or muscle stem cells) are particularly important in the repair and maintenance of muscle fibers. Satellite cells can perform muscle repair or hypertrophy or return to an inactive state [118]. In an adult individual, the number of satellite cells (SCs) present in skeletal muscle tissue remains reasonably constant, providing a safety reservoir that can be used in situations of need [13]. SCs are myogenic progenitors, endowed with a self-regeneration capacity, favoring the constant maintenance of their population and or even its increase when necessary [119]. In a study carried out with elderly rodents, it was observed that increased ROS generation results in an irreversible pre-senescent state together with a loss of normal quiescence (generally reversible) in some satellite cells. This condition definitively affects the self-regenerative capacity of these cells, as well as their activation for a repair process, in the case of an injury [120]. The lack of regenerative capacity in sarcopenic muscle, in addition to being a hallmark of aging, is one of the main causes of loss of independence in the elderly [121,122,123,124].

Senescence, a state of stable growth arrest that occurs in response to genomic, proteomic, metabolic, or replicative stress, is primarily the fate of proliferating cells. The mechanisms involved in programming senescence are triggered by cell cycle inhibitory proteins, including p16^Ink4a^ and p21^Cip1^, which antagonize the actions of cyclin-dependent kinases to ultimately halt cell proliferation [125,126,127]. In this sense, senescence is considered a mechanism of protection and tumor suppression [128]. However, with advancing age, senescent cells accumulate, presumably due to their resistance to apoptosis and inefficient removal by the immune system [5,129,130,131]. The depletion of mitotically active progenitor cells, combined with the pro-inflammatory microenvironment, generated by the abundant secretion of cytokines, chemokines, matrix remodeling proteins, and growth factors, due to the accumulation of senescent cells, results in compromised regeneration and the consequent deterioration and fibrosis of the tissues of older organisms [127,131,132]. Consistently, increased levels of p16 mRNA and the SASP factors IL-1α, IL-1β, IL-6, and TNF-α were observed in the skeletal muscle of aged mice compared to the younger individuals, along with the morphological features of skeletal muscle aging, such as reduced fiber size and increased frequency of centrally nucleated fibers, and changes in several markers of cellular senescence [133].

Furthermore, advancing age is associated with aberrant signaling from fibroadipogenic progenitors (FAPs), culminating in pro-fibrotic behavior and the fibrosis of skeletal muscle, leading to compromised satellite cell function [134]. It remains unclear whether metabolic stress or other forms of age-related damage lead to the differentiation of FAP into adipocytes and, in turn, cause the accumulation of intermuscular fat [135]. Interestingly, it has been shown that fat accumulation in skeletal muscle after an injury increases with aging [136]. However, a later study reported that young mice activate stronger ectopic adipogenesis than older mice during tissue repair [134]. These reports corroborate the findings in adipose tissue, in which mesenchymal progenitor cells showed a loss of adipogenic potential with advancing age, while the clearance of senescent adipogenic progenitor cells restored metabolic health and adipogenesis [102,137].

Similar to adipose tissue, skeletal muscle macrophages are the main population of immune system cells. Macrophages play a critical role in the body’s response to skeletal muscle injury, infiltrating tissue to remove cellular debris and secreting pro-inflammatory mediators and growth factors to promote healing and recruit additional immune cells [138]. Macrophages are also involved in the activation and differentiation of SCs for a successful skeletal muscle repair [139,140]. However, the role of macrophages in sarcopenia is currently poorly understood. In humans, the M1 macrophages in skeletal muscle decrease with aging, whereas the M2 macrophages increase [141]. The secretome released by M1 macrophages reduces the adipogenic potential of FAPs, while the secretome of the M2 macrophages promotes the stimulation of this process, suggesting that the age-related increase in the M2 subtype may contribute to the increase in intermuscular fat that comes with aging [142,143] (Figure 3).

## 5. Impacts of Senescence on Body Composition and Disease Development

The changes in body composition that occur during the aging process are associated with factors such as weight and other age-related conditions, in addition to physiological and behavioral determinants. Such circumstances are complex and interactive and contribute to additional changes in body composition and a decline in general health conditions [14]. In this context, dysfunctional adipose tissue plays a critical role in lipid redistribution and the establishment of a pro-inflammatory phenotype observed with advancing age [144,145,146]. These conditions promote an increased risk of developing metabolic disorders, such as insulin resistance, impaired glucose tolerance, and dyslipidemia, ultimately resulting in the development of diabetes and cardiovascular diseases [147,148].

Moreover, the reduction in muscle mass observed with aging, accompanied by fat infiltration and the connective tissue within the muscle, is linked to the pathogenesis of sarcopenia, which can result in extreme conditions like frailty syndrome [149,150,151].

In the following sections, we will discuss how changes in body composition can be affected by the process of senescence in aging and lead to the onset of diseases.

### 5.1. Diabetes

Similar to other aging-related diseases, the development and accumulation of senescent cells are underlying features of diabetes and obesity. Senescent cells are involved in tissue imbalances and the development of the metabolic disorders associated with obesity and diabetes. They directly affect the function of pancreatic β-cells and promote tissue damage through a SASP, contributing significantly to adipose tissue dysfunction and the pathogenesis of type 1 (T1D) and type 2 (T2D) diabetes [152,153].

The central role of insulin resistance and decreased insulin secretion in the pathophysiology of T2DM is well established [154,155]. Insulin resistance occurs at the expense of impaired insulin signaling in peripheral tissues such as liver, muscle, and fat, and can be accelerated by the presence of obesity and accumulation of dysfunctional adipose tissues, followed by a compensatory increase in insulin secretion [154,156]. The evolution of these imbalances promotes a decline in the function of pancreatic β cells and results in the establishment of diabetes mellitus in prone individuals [157,158,159]. Consistent with the notion that the quantities and functions of adipose tissue are affected by aging and by factors such as caloric intake and physical activity, among other aspects of health status, and that aging is associated with increased insulin resistance, studies indicate that cellular senescence in adipose tissue contributes significantly to the initiation and progression of imbalances in insulin action [74,155,160,161,162].

Additionally, glucose tolerance is also affected by age in patients with DM2, reflecting a progressive decline in the response of β cells to glucose stimulation and the reduced sensitivity of peripheral tissues to insulin. Preceding hyperglycemic changes, increased insulin secretion can be observed as a compensatory response of β cells to increased metabolic demands. However, this compensation becomes limited by the age-related decline in β-cell proliferation observed in both rodents and humans [160,163,164]. This impaired proliferation in response to an increased insulin release may partly arise from the accumulation of senescent β-cells [165,166].

Complementary investigations also reported that other indicators of the presence of senescent β-cells can be identified from the reduction in β-cell proliferation and increased expression of senescence markers, as demonstrated in a study carried out in a murine model of DM2 induced by a high-fat diet, in addition to a significant reduction in the length of telomeres in β cells from human individuals with DM2 [160,167,168,169]. It has also been reported that factors such as elevated glucose levels may contribute to the dysfunction of telomerase activity in β-cells through increased oxidative stress in these cells [170]. Furthermore, another study showed that the shortening of telomeres in pancreatic β cells caused by mutations in TERT in murine models resulted in fasting hyperglycemia and an increased expression of p16Ink4a in pancreatic islets as well as the impairment of crucial events for insulin exocytosis, such as such as the hyperpolarization of the mitochondrial membrane and Ca2+ mobilization in these cells [171].

Furthermore, a new human study reported that there was an increase in the number of senescent cells in islets isolated from the pancreas of healthy elderly people compared to younger individuals. Furthermore, the same study observed that this number is even higher in islets from the patients with T2DM compared to the non-diabetic individuals, suggesting that β-cell senescence may contribute to the pathogenesis of T2DM [172]. Complementary data in mice, published by this same research group, showed that senescent β-cells showed an inhibition of the expression of β-cell-identifying genes as well as a simultaneous increase in the expression of senescence markers and a SASP [173]. Senescent β-cells increase basal insulin secretion and are transcriptionally rewired, exhibiting a decreased expression of the genes required for cellular depolarization, incretin signaling, and insulin granule production, all of which are required for insulin secretion [166,173].

The presence of senescent β-cells in the pancreas of T1D patients compared to non-diabetic individuals has also been described [174]. It has been reported that in addition to the progressive loss of healthy β-cells in T1Ds, senescent β-cells accumulate during the development of T1Ds in humans and in a non-obese diabetic (NOD) mouse model of T1Ds [174,175]. The senescent β-cells in the NOD mice were identified through the overexpression of p16^Ink4a^ and p21, the activation of the Bcl-2-mediated DNA damage response, increased SA-βgal activity, and the expression of the SASP factors primarily involving IL-6 [174]. The presence of a persistent DNA damage response and the selective upregulation of Bcl-2 and SASP content generally differentiated the stress-induced senescence occurring in the β-cells of the NOD mice caused by natural aging from β-cell senescence as reported in the model’s mice with T2DM and maturity-onset diabetes of the young (MODY) [172,176,177]. Notably, in the NOD mice, the elimination of senescent β-cells with senolytics or the suppression of SASPs halts disease progression. Such findings highlight senescence as a promising new therapeutic target during T1D progression [174,178].

In addition to exclusive cytokines and adipokines, adipose tissue is also a source of FGF21. Although the liver is considered the primary source, adipocytes have also been shown to produce FGF21 to varying degrees in response to various stimuli. FGF21 levels are associated with obesity and diabetes, and more recently with an increased risk of developing cardiovascular disease (CVD) [179,180,181,182]. In this sense, a study showed that diabetic individuals diagnosed with CVD had high levels of FGF21, suggesting an important role for FGF21 in atherosclerosis accelerated by diabetes [183]. In addition, a positive association between FGF21 levels, hypertension, and triglyceride levels was reported, as well as a negative association between this marker and HDL cholesterol levels [184]. Another study revealed that plasma FGF21 levels are associated with increased pericardial fat deposition, suggesting that ectopic fat could be a source of FGF21 in metabolic diseases. In contrast to the effects of physiological FGF21, studies have demonstrated that the pharmacological administration of FGF21 in humans and primates promotes a reduction in glycemic, insulin, triglyceride, and LDL cholesterol levels while increasing HDL cholesterol [185,186,187,188]. In light of such claims, further studies are needed to clarify how FGF21 levels may affect factors related to an increased CVD risk, as well as to discern whether adipocyte- or liver-derived FGF21 contributes to these observed effects.

### 5.2. Cardiovascular Diseases

Studies support the possibility that senescent cells contribute to age-related systemic inflammation [137,189,190,191,192]. In this context, a study showed that senescent cells accumulate in aged adipose tissues as well as in the other tissues of rodents and humans [193]. Furthermore, it was demonstrated that senescent preadipocytes developed a SASP capable of inducing inflammation in adjacent healthy cells as well as in the entire adipose tissue, and that senescent cells overexpressing p^16Ink4a^ were the main cause of increased IL-6 levels in the stromal vascular fraction of adipose tissue from older progeroid mice [28,103]. Thus, adipose tissue is an important source of circulating inflammatory cytokines in elderly populations, in addition to being considered the largest organ in most humans, making it a good model for studying cellular senescence and the SASP [102,110,194,195,196,197].

Adipose tissue is capable of promoting systemic inflammation as it can act as an important source of inflammatory mediators [198]. It is known that chronic inflammation causes a significant increase in the risk of developing cardiovascular diseases (CVDs). The secretion of pro-inflammatory cytokines by inflamed adipose tissue can cause changes in the composition of perivascular adipose tissue and accelerate atherosclerosis [199]. Furthermore, among the various factors that contribute to the development of cellular senescence during the progression of atherosclerosis, dyslipidemia appears to play a fundamental role [200]. Dyslipidemia is characterized by a subset of lipid metabolism disorders involving abnormally elevated plasma levels or the functional impairment of lipids or lipoproteins. Evidence indicates that low-density lipoprotein (LDL) and its modifications provoke pro-atherogenic effects through the induction of vascular cell senescence, while the elimination of these senescent cells stabilizes the fibrous cap [200,201,202,203]. Furthermore, remnant lipoproteins (RLPs) and fatty acids (FAs), which are hydrolyzed products of triglyceride-rich lipoproteins (TRLs), accelerate cellular senescence in atherosclerosis [204,205,206,207]. In addition, high-density lipoprotein (HDL), although associated with anti-senescence effects, when in high concentrations or dysfunctional, can result in cellular senescence, suggesting that dyslipidemia may be a key factor contributing to linking cellular senescence to atherosclerosis [208,209,210].

Furthermore, adipose tissue-derived cytokines, such as IL-1β and TNF, induce the expression of endothelial cell adhesion molecules, resulting in the exacerbation of the vascular inflammatory process [211]. Furthermore, exosomes derived from inflamed visceral adipose tissue have been shown to promote the pro-inflammatory polarization of M1 macrophages and atherosclerosis [212].

Therefore, aging can be considered an independent risk factor for increased morbidity and mortality from atherosclerosis [213,214,215,216]. Interestingly, cellular senescence and atherosclerosis share common causal factors, such as hyperlipidemia, hypertension, diabetes, and obesity; in addition, cellular senescence is itself considered an important factor in the development of atherosclerosis [200,215,217,218,219,220]. Additional investigations also indicate the presence of several types of senescent cells in atherosclerotic arteries, including endothelial cells, vascular smooth muscle cells, and macrophages [218,219,221,222,223]. Additionally, evidence has also indicated that the senescence of other cell types, such as mesenchymal stem cells and endothelial progenitor cells derived from adipose tissue, also participate in the pathophysiological process of atherosclerosis [224,225,226,227].

### 5.3. Sarcopenia

Dysfunctional adipose tissue promotes lipid redistribution and can result in age-related functional impairments. As mentioned previously, aging can promote increased lipid infiltration in skeletal muscle. The deposition of these lipids in different regions of this tissue, such as intermuscular, intramuscular (within the muscle, but between the fibers), and intramyocellular, has been associated with functional declines in older adults [146,228,229]. An inverse relationship between the number and size of lipid droplets in myofibers and the loss of contraction speed and energy generation has been reported in an in vitro assay using individual muscle fibers from older adults. This same study suggested that the occurrence of this event probably contributes to the reduction in muscle strength and power development observed in the same individuals in vivo [230]. A strong relationship is consistently established between the amount and location of fat and muscle function, both at a cellular and systemic level, in older adults, which may result in impairments in functional capacity and mobility [117,229,231].

Deposition of intramyocellular lipids results in a lipotoxic effect, which induces and exacerbates mitochondrial dysfunction, increases oxidative stress, and contributes to insulin resistance and inflammation [232]. An important reason is that local muscle insulin resistance leads to a reduced lipid uptake and increases the local concentrations of free fatty acids (FFAs), thus worsening local hyperlipidemia and the inflammation of intermuscular adipose tissue, further increasing the secretion of pro-inflammatory factors in adipose and muscle tissues [233]. The inflammatory process induced by a SASP with aging can result in a systemic propagation of this metabolic dysfunction, resulting in lipid dysfunction and senescence in adipose depots, muscles, and other tissues [234].

The accumulation of senescent cells in muscle tissue is involved in the progression of sarcopenia due to the pro-inflammatory stimulus, resulting in increased protein degradation and the consequent thinning of muscle fibers [133,235,236]. Furthermore, cellular senescence affects the functionality and number of satellite cells (SCs). SCs are muscle-specific stem cells and play a crucial role in muscle fiber regeneration since as they transition from their normal quiescent state with a low metabolic rate to an active state to proliferate, they differentiate into new muscle fibers [120,237]. Thus, subsequent cellular senescence could limit the regenerative capacity and maintenance of SCs, since p16^INK4a^ overexpression is the main inducer of cell cycle arrest in cellular senescence [120]. On the other hand, the SASP significantly contributes to sarcopenia through inflammatory stress, in which elevated levels of IL-6 damage muscle integrity and function, resulting in skeletal muscle degradation and atrophy [238]. In this sense, the findings of the in vitro and in vivo models suggest that pro-inflammatory cytokines are involved in both the process of muscle atrophy and functional decline. For example, in a study carried out in elderly animals, it was reported that the levels of TNF-α, IL-1β, and IL-6 detected in skeletal muscle were able to predict the existence of declines in grip strength, locomotor capacity, and resistance in this model [193,239]. Similarly, other investigations have reported that chronic inflammatory disease states can result in intense catabolic effects on various tissues, also leading to skeletal muscle atrophy [240,241]. Furthermore, the chronic activation of NF-κB can promote muscle atrophy, while the targeted ablation of an NF-κB activator can improve muscle strength, mass, and regeneration [242,243]. In addition, chronic and systemic inflammation can result in disrupting anabolic signaling, since IL-6 and TNF-α are associated with the inhibition of insulin signaling, IGF-1 and erythropoietin production, as well as post-inflammatory protein synthesis postprandial or after exercise [244]. The main findings on the involvement of inflammatory cytokines in functional decline and sarcopenia may come from a study that used a fragile mouse model that was developed from the targeted depletion of IL-10, an important anti-inflammatory cytokine. These animals showed a considerable increase in the basal levels of circulating cytokines, including IL-1β, IL-6, and TNF-α, in addition to muscle weakness, changes in their skeletal muscle gene expression profile, endocrine dysfunctions, IGF-1 dysregulation, and increased mortality rates [245]. Given the evident damage caused by cellular senescence in the process of skeletal muscle degeneration, the need to find new therapeutic strategies to mitigate cellular senescence and the impact of a SASP is clear.

### 5.4. Cancer

Cancer can arise at the site of inflammation, and a pro-inflammatory microenvironment is an essential component of cancer. Chronic inflammation can initiate cancer, promote its progression and support its metastatic spread [246]. Inflammation of adipose tissue may also be the cause of cancer [12].

Many tumors develop incorporated in or immediately adjacent to adipose tissue. In addition to endocrine functions, evidence shows the important paracrine effects of adipose tissue that directly affect tumor development, growth, and progression. This interaction is best characterized in breast cancer, where the mammary glands are embedded in the fatty tissue of the breast, but also exists in pancreatic, kidney, melanoma, and prostate cancers, as well as in the multiple myelomas that accumulate in bone marrow. In turn, adipose tissue also responds to tumor-derived factors, with changes in adipokines, hormones, cellular metabolism, ECM deposition, and vascularization, supporting the growth and progression of cancers [247].

Although investigations associated with the impacts of aged adipose tissue on the development of cancer are still scarce, such a relationship has already been characterized in the adipose tissue from obese individuals and, since both conditions carry with them the presence of common factors including the inflammatory process which plays a fundamental role, these data offer insight into the possible contribution of this tissue to the development of cancer with advancing age. In this sense, studies aimed at understanding the factors associated with the development and progression of tumors in obese individuals have pointed to the existence of an association between the altered tumor microenvironment in an obese state and local and systemic inflammation linked to the established and emerging characteristics of cancer [248,249]. Chronic inflammation occurs in response to the loss of tissue homeostasis resulting in an aberrantly prolonged and unbalanced protective response, being considered an underlying factor implicated in the activation of tumorigenesis mechanisms [248,250]. Furthermore, inflammatory stimuli or somatic mutations can amplify the activity of the intracellular signaling pathways involved in inflammation such as NF-kB [251,252]. The activation of NF-kB results in the stimulation of the expression of the genes involved in pro-inflammatory signaling, and modulation of cell proliferation and the survival pathways, which can result in the dysregulation of the cellular senescence mechanisms [104]. These changes have already been demonstrated to be involved in the epithelial–mesenchymal transition (EMT), a process by which cancer cells become more invasive and acquire metastatic potential and genomic instability due to a deficiency in the DNA repair mechanisms and the consequent increase in damage to the molecule [253,254].

As previously mentioned, NF-κB is an important regulator involved in the cellular senescence that occurs in adipose tissue, as well as in the development of the SASP [12,255]. Evidently, NF-κB is a mediator involved in the tumor-promoting mechanisms of both early-stage pro-inflammatory tumor-associated macrophages (TAMs) and late-stage immunosuppressive TAMs [256,257]. Furthermore, another interesting finding provided clear evidence of the similarities between adipose tissue macrophages and TAMs [258]. Complementary studies showed that adipose tissue macrophages from obese individuals expressed the protumor genes commonly identified in TAMs, including cytokines, chemokines, proteases, as well as growth and angiogenic factors. Indeed, many of these tumor-promoting genes, including VEGF-C and CXCL12, two known targets of NF-κB signaling, were expressed in similar or greater proportions in obese adipose tissue macrophages compared to TAMs [258,259,260]. Such evidence indicates that chronically activated NF-κB signaling and the dysregulated immune responses arising from these stimuli are probably an important link between adipose tissue macrophages and TAMs.

Thus, the establishment of an association between adipose tissue and the development of cancer, favored by the presence of a low-grade chronic inflammatory process at the tissue and systemic levels, common to conditions such as obesity, and present with the advancement of age, can strongly contribute to the increased risk of the development and progression cancer at this stage of life.

## 6. Polyphenols, Body Composition, and Senescence

The applicability of substances with senotherapeutic properties has recently emerged as a promising proposal to extend life expectancy and quality, in addition to preventing the development of chronic diseases. In this sense, dietary phytochemicals can promote longevity by modulating the mechanisms associated with metabolic activities and cellular processes in a similar way to other anti-aging strategies, such as calorie restriction, intermittent fasting, and physical exercise [261]. The benefits of the dietary phytochemicals derived from fruits and vegetables can be attributed to the activation of the stress resistance mechanisms. Some natural senolytic compounds and pharmacological agents exert anti-senescence effects through an interaction with the molecular targets associated with aging [262]. The senolytics of natural origin, although they may be less effective compared to synthetic senolytics, have the advantage of a low toxicity and may be promising candidates for a translation into clinical settings or for the development of more specific and potent senolytics [152].

Considering the known efficacy of polyphenols as potent antioxidants, there has been speculation about their ability to prevent cellular senescence and slow down the aging process [18]. In this context, polyphenols have been reported to inhibit senescence by suppressing β-galactosidase activity as well as the expression of senescence-associated targets such as p16Ink4a, p21Cip1, and p53 in several models [263,264,265,266,267]. Furthermore, it has also been demonstrated that the establishment of a SASP in senescent cells is modulated by polyphenols [268,269,270,271,272,273,274,275,276] (Table 1).

A therapeutic approach that explores the use of polyphenols with senolytic activity emerged in 2015, with the use of quercetin, which, in studies, has been shown to prolong the life of elderly rats, as well as improve muscle strength and physical capacity during exercise [129,277]. Quercetin also has anti-inflammatory and immunomodulatory actions and has demonstrated protective effects against dexamethasone-induced skeletal muscle atrophy, regulating the protein-Bx/Bcl-2 ratio and abnormal mitochondrial membrane potential (ΔΨm), leading to the suppression of apoptosis [278,279]. This polyphenol has been reported to exert positive metabolic effects on blood pressure, HDL cholesterol, and triglycerides, as well as exert pleiotropic effects on senescent cells [280,281]. Furthermore, quercetin may have an inhibitory effect on adipogenesis [282]. In clinical trials, quercetin is often combined with dasatinib, a tyrosine kinase inhibitor. This combination induces apoptosis in adipocytes and increases the senolytic effects [283,284]. In another investigation evaluating the senolytic potential of quercetin, it was reported that pre-adipocytes and adipocytes exposed to H_2_O_2_ acquired a senescent profile, identified by the increased activity of β-galactosidase (SA-β-gal) and p21, as well as the increased expression of pro-inflammatory cytokines and ROS activation. However, after treatment with quercetin there was a significant reduction in the number of cells positive for SA-β-gal, concomitant with the inhibition of ROS and pro-inflammatory cytokines. Furthermore, the inhibition of miRNA-155-5p, possibly through the modulation of NF-κB and SIRT-1, was associated with the anti-senescence effects of quercetin in these cells [285]. In addition, data from a recent clinical trial carried out with diabetic individuals provided evidence that senolytic agents are capable of reducing the accumulation of senescent cells in human tissues. This study reported that the combined treatment of dasatinib and quercetin was able to reduce the accumulation of senescent cells in adipose tissue within 11 days, by decreasing the number of cells expressing p16^Ink4a^ and p21, inhibiting SA-β-gal activity, as well as imbuing the SASP and adipocyte progenitors with limited replicative potential [263]. In another study, quercetin, as well as other calorie restriction mimetics such as curcumin and EGCG, was reported to protect the heart from aging and heart failure [286].

Another polyphenol, EGCG, has been shown to act as an mTOR inhibitor, SASP modulator, and a potential senolytic agent. In this study, EGCG significantly inhibited PI3K/Akt/mTOR and AMPK pathway signaling, along with the suppression of the ROS, iNOS, Cox-2, NF-κB, SASP, and p53-mediated cell cycle inhibition in 3T3 preadipocytes. Furthermore, the EGCG treatment was also able to promote the apoptosis of senescent cells by suppressing the accumulation of the antiapoptotic protein Bcl-2 [287]. A similar study also demonstrated a potent anti-SASP effect in the 3T3-L1 cells, achieved with this polyphenol through a significant inhibition of IL-6 expression [288]. Another investigation, in turn, showed that the anti-senescence effects of EGCG can also be observed at a macroscopic level, as its supplementation reduced age-related sarcopenia in a murine model [264].

Still, pre-clinical studies exploring the role of polyphenols in mitigating cellular senescence and its effects are still scarce. A recent randomized, double-blind, crossover, placebo-controlled study suggested that supplementation with ginsenoside Rg1, administered 1 h before high-intensity cycling, was effective in eliminating senescent cells as well as at improving high-intensity endurance performance in human skeletal muscle [289].

Curcumin has demonstrated anti-aging effects in several experimental approaches; however, there is only limited data showing its ability to modulate cellular senescence. In this context, a study used curcumin in a murine model of chemically induced DM1, characterized by an impairment of the endothelial progenitor cells (EPCs). The results revealed that curcumin application was able to significantly improve blood circulation and increased capillary density in the ischemic hind limbs in this model. The in vitro data from this study showed that the angiogenesis, migration, and proliferation capabilities of EPCs and the number of EPCs positive for SA-β-gal returned to non-pathological levels after treatment [290]. Complementarily, in another study carried out in atherosclerotic rats, curcumin decreased SA-β-gal activity in the aorta, as well as the MCP-1 levels in the serum of the treated animals [291]. In another investigation, curcumin increased the survival of rat mesenchymal stem cells and indirectly decreased the cell population doubling time, indicating that this polyphenol may influence replicative senescence [292]. Furthermore, a recent study performed on aged mice suggested a possible muscle-specific response to curcumin treatment. In particular, chronic systemic administration of this polyphenol was able to significantly reduce spontaneous mortality during senescence, effectively combating pre-sarcopenia as well as significantly attenuating sarcopenia by improving the commitment and recruitment of satellite cells [293]. An investigation carried out in primary cultures of embryonic fibroblasts from prematurely aged rats, curcumin promoted a slightly smaller reduction in the number of senescent cells compared to fisetin, which is considered a potent senolytic agent [294]. In the same study, the administration of fisetin to Ercc1−/− progeroid mice was able to inhibit the SASP in all tissues, in addition to significantly reducing the number of cells expressing p16Ink4a in the adipose tissue of this model [294].

Fisetin, a member of the flavonoid family, has shown potential as a senolytic agent [295,296,297]. Another study with this polyphenol revealed that human adipose tissue-derived primary stem cells (ADSCs), a population of fat-resident mesenchymal stem cells (MSCs) commonly used in regenerative medicine applications, express common markers of cellular senescence, including increased SA-β-gal activity, ROS activation, and the presence of heterochromatin foci associated with senescence. However, fisetin administration acted in a dose-dependent manner to selectively attenuate these markers of senescence while preserving the differentiation potential of the expanded ADSCs [298]. Fisetin treatment has also been shown to restore muscle stem cell function in dystrophic and progeroid mice, exerting effects on the senescent cells of diseased or aged muscle tissues [299,300].

Resveratrol, in turn, is well known for its antioxidant and anticancer effects [301,302]. Furthermore, it has anti-neuroinflammatory properties, protecting against memory impairment in a rat model of cerebral palsy [303]. Although studies related to the impacts of its senolytic effects on body composition are scarce, resveratrol has been reported to significantly inhibit the SASP factors such as IL-1β, IL-8, and TNF-α, mainly through its inhibition of NF-κB activation [304]. Such data could be useful in studies related to the diseases associated with aging resulting from metabolic imbalances associated with the changes in body composition during aging, such as sarcopenia, diabetes, dyslipidemia, and cardiovascular diseases, among others.

Proanthocyanidins (condensed tannins) have been linked to health benefits, combating the imbalances related to the development of cardiovascular disease, certain types of cancer, diabetes, and inflammation [305]. For example, it has been shown in a study that type B proanthocyanidins, derived from the cinnamon species *Cinnamomum cassia*, regulate the accumulation of lipids in adipose tissue and the liver, while type A proanthocyanidins extracted from *Cinnamomum tamala* increase the concentration of insulin in the blood and in the pancreas [306]. Furthermore, studies revealed the anti-inflammatory potential of proanthocyanidins, through the significant inhibition of pro-inflammatory mediators, such as NADPH oxidase, inducible nitric oxide synthase, COX-2, IL-6, and TNF-α in vitro and in vivo [307,308,309,310,311].

Ellagic acid has broad therapeutic effects in both diabetes and muscle injuries due to its multiple biological and pharmacological properties [312,313]. Although studies aimed at its anti-senescence activity are scarce, the modulation potential of this substance on body composition tissues was observed in a study. In this study, it was demonstrated that ellagic acid supplementation promoted improvements in the size and weight of muscle fibers and grip strength in a murine model of diabetes with streptozotocin-induced muscle atrophy [314]. The data showed a significant reduction in the expression of the genes associated with muscular atrophy, such as Atrogin-1 and MuRF-1, as well as the promotion of an improvement in the mitochondrial dysfunction in the skeletal muscle tissue of this model. Furthermore, the treatment was able to inhibit apoptosis through BAX gene inhibition and Bcl-2 stimulation [315].

In summary, polyphenols have been identified as promising candidates for slowing down the aging process. Its modes of action are diverse and can impact cell longevity, SA-β-gal activity, as well as the other mechanisms associated with senescence, directly or indirectly [152,294,295,316,317] (Figure 4). Evidence from animal studies has shown that polyphenols exert their anti-senescence effects mainly through senomorphic properties, although some also exhibit senolytic properties [277,283,316].

**Table 1 nutrients-16-03621-t001:** Benefits of polyphenols in preventing cellular senescence during aging in body composition tissues.

Polyphenol	Model	Intervention/Treatment	Results	Reference
Quercetin	C2C12 cells	Quercetin (25, 50, 75, and 100 μM) for 24 h	Downregulation of Bax and ROS and the reversal of the ΔΨm imbalance	[279]
	3T3-L1 cells	Quercetin (0.1–10 µM) for 24 h	Reduced C/EBPβ, SREBP1, and PPARγ gene expression; reduced LPL, FAS expression and triacylglycerol content.	[282]
	3T3-L1 senescent cells	Quercetin 20 μM	Reduction of SA-β-gal activity, ROS, and inflammatory cytokines; inhibition of miR-155-5p expression; NF-κB and p65 downregulation and SIRT-1 upregulation.	[285]
EGCG	3T3-L1 senescent cells	EGCG (50 and 100 μM)	Downregulation of PI3K/Akt/mTOR and AMPK signaling; suppression of ROS, iNOS, Cox-2, NF-κB, SASP, and p53 mediated cell cycle inhibition; suppression of anti-apoptotic protein Bcl-2 accumulation in senescent cells.	[287]
	3T3-L1 senescent cells	EGCG 50 μM	Reduced IL-6 and CDKN1a expression levels; NRF2 and SIRT3 activation.	[288]
	SAMP8 mice (32 weeks old) and late passage C2C12 cells	Diet containing without or with 0.32% EGCG for 8 weeks	Increased miRNA-486-5p expression in both aged SAMP8 mice and C2C12 cells; AKT phosphorylation stimulation and inhibition of FoxO1; MuRF1 and Atrogin-1 in vivo and in vitro; increased Myostatin expression in C2C12 cells.	[264]
		Cells treatment with EGCG (50 μM) for 24 h.		
Ginsenoside Rg1	12 young men (age 21 ± 0.2 years)	Rg1 (5 mg) supplementations 1 h prior to a high-intensity cycling (70% VO2max)	Suppression of SA-β-gal activity in exercised muscle samples.	[289]
Curcumin	Male C57/B6 mice (4–6 weeks old)	1000 mg/kg curcumin once a day for 14 days	Overexpression of VEGF-A and Ang-1 in EPCs and number of senescent EPCs reduced.	[290]
	C57BL/6J mice (8 weeks old)	Curcumin (0.1%) in HFD until 80 weeks old	Reduction of SA-β-Gal activity and MCP-1 expression; enhanced of HO-1 activity; suppression of urinary 8OHdG and superoxide production.	[291]
	rADSC cells	Curcumin (1 and 5 µM) for 48 h	Suppression of SA-ß-gal activity; increased TERT expression levels.	[292]
	C57BL6J and C57BL10ScSn (18 months-old)	Curcumin (120 μg/kg) injected subcutaneously (every 6th day, for 6 months)	Preserved type-1 myofiber size and increased type-2A one in soleus; increased MyoD-positive satellite cells from old hindlimb muscles.	[293]
Fisetin	Progeroid mice (p16^+/Luc^;Ercc1^−/∆^); murine embryonic fibroblasts (Ercc1^−/−^ MEFs) and human fibroblasts (IMR90)	Fisetin-enriched diet (500 mg/kg)	Reduction of SA-β-gal activity; expression of p16^Ink4a^, p21^Cip1^ and the SASP factors in vitro and in vivo in white adipose tissue.	[294]
		Cells treatment with fisetin (1–15 μ M) for 48 h		
	HUVEC cells	Fisetin (1–100 μM) for 72 h	Induces apoptosis in senescent cells.	[295]
	rADSCs cells	Fisetin (25, 50 or 100 µM) for 24 h	Reduction of ROS, SA-β-gal activity, and senescence-associated heterochromatin foci.	[298]
Resveratrol	Male Wistar rats	Resveratrol (10 mg/kg, 0.1 mL/100 g)	Reduction in gene expression of IL-6, TNF-α and increase in Creb-1 levels; reduction of activated microglia and increase in cell proliferation.	[303]
	Annual fish *N. guentheri*	Resveratrol-enriched diet (200 μg/g food) until 12 months old	Inhibition of NF-κB by decreasing RelA/p65, Ac-RelA/p65 and p-IκBα levels; reduced SA-β-gal activity; SASP-associated pro-inflammatory cytokines IL-8 and TNFα and increases anti-inflammatory cytokine IL-10; increased SIRT1 expression.	[304]
Proanthocyanidins	*Female Zucker fa*/*fa* rats (5 weeks old)	GSPE (35 mg/kg/day) for 10 weeks	Inhibition of iNos and IL-6 expression and increased Adiponectin levels in adipocytes.	[307]
	Male ICR mice (6 weeks old)	Procyanidin B2 by gavage (25, 50, and 100 mg/kg) for 7 days	Decreased TNF-α, IL-1β, COX-2, and iNOS expression; inhibited the translocation of NF-κB/p65 from the cytosol to the nuclear fraction in mouse liver; inhibited CCl4-induced hepatocyte apoptosis; suppressed the upregulation of Bax expression and restored the downregulation of Bcl-xL expression.	[308]
	Wistar female rats (11 weeks old)	GSPE (75 and 200 mg/kg/day)	Downregulated the genes Il-6 and iNos; decreased the glutathione ratio (GSSG/total glutathione).	[309]
	Crossbreed male pigs (Landrace × Yorkshire)	Procyanidin-enriched diet (0.01%, 0.02%, and 0.04%) for 4 weeks	Reduction in the levels of IL-1β, IL-6, and TNF-α at 4 h after LPS challenge.	[310]
	THP-1-macrophages	Procyanidin B2 (10 μM) for 1–4 h	Inhibition of inflammasome activation includes the inactivation of the NF-κB/p65 nuclear expression; repression of COX2, iNOS, IL-6, IL-1β, and NO; decreases NLRP3 and caspase-1 activation.	[311]
Ellagic acid	Male C57BL/6 mice (6 to 9 weeks old)	Ellagic acid (5, 50, and 100 mg/kg P.O) for 42 days	Reduction of lipid peroxidation, ROS, and Sirt3 levels; increased antioxidant capacity, GSH/GSSG ratio, and mitochondrial activity.	[313]
	Male ICR mice (8 weeks old)	Ellagic acid (100 mg/kg/day) for 8 weeks	Enhanced fiber size and weight of gastrocnemius, and grip strength; decreased Atrogin-1 and MuRF-1 expressions; increased NRF-1 and PGC-1α expressions to alleviate mitochondrial disorder; CHOP and GRP-87 levels to relieve ER stress; inhibited BAX expressions and enhanced Bcl-2 expressions to mitigate apoptosis.	[315]

## 7. Clinical Evidence

Recent research suggests that dietary polyphenols may offer promising benefits for extending lifespans and mitigating age-related diseases by targeting cellular senescence and the senescence-associated secretory phenotype (SASP). These compounds have shown potential in reducing the markers of aging and improving regenerative functions in pre-clinical studies.

The clinical studies available in the clinicaltrials.gov database show that various research groups across different regions of the world have been dedicated to exploring the potential of polyphenols in relation to multiple aspects of aging (Table 2).

Previously published studies endorsed the benefits of these compounds in various contexts associated with aging. For example, a South Korean study explored the prolonged ingestion of a cocoa flavonoid-rich product on the aspects associated with photo-aging, and the results of this investigation showed that in moderately photo-aged women, the regular consumption of cocoa flavanols had positive effects on facial wrinkles and elasticity, suggesting that cocoa flavanol supplementation may contribute to preventing the progression of photo-aging (NCT02060097) [318]. In another study, the effects of red beet juice were investigated on endothelial dysfunction induced by high-fat meals and cardiometabolic disorders in a randomized, placebo-controlled clinical trial conducted in the United States (NCT02949115). This study reported that supplementation was able to modulate postprandial endothelial function and other cardiometabolic responses in overweight/obese individuals over 50 years old [319]. A clinical study published by researchers in the United Kingdom on the impacts of cranberry consumption on the microbiome and brain for healthy aging showed that daily cranberry supplementation (equivalent to a small cup of cranberries) over a 12-week period improves episodic memory performance and neural functioning, providing a basis for future investigations to determine its effectiveness in the context of neurological diseases (NCT03679533) [320]. Additionally, a study published by researchers in the United States on the effects of pomegranate juice supplementation on aging, particularly on memory in the elderly, showed that daily consumption of pomegranate juice can stabilize the ability to learn visual information over a 12-month period (NCT02093130) [321]. In another study conducted by researchers in the United States, it was shown that daily consumption of blueberries for 8 weeks was able to reduce blood pressure and arterial stiffness, effects that were partially attributed to increased nitric oxide production (NCT01686282) [322].

Moreover, the oral bioavailability of curcuminoids in healthy humans has been evaluated (NCT01982734). This study sought new strategies to increase the potency of nutraceuticals with low oral bioavailability and their application in new functional foods for the optimal protection of the aging brain. The oral bioavailability of curcumin is low due to its poor aqueous solubility, limited gastrointestinal absorption, rapid metabolism, and excretion. In this randomized crossover study, the simultaneous application of phytochemicals and micellar solubilization, both separately and together, were used as strategies to increase curcumin absorption in the body. It was demonstrated that this bioavailability is markedly increased by micellar solubilization but is not improved by the simultaneous intake of sesamin, ferulic acid, naringenin, and xanthohumol [323].

Further investigations aimed at confirming the efficacy of polyphenol consumption in clinical trials are currently underway and will be described below. It is important to note that, although some of these studies have already been completed, they have yet to be published. For example, an ongoing study conducted in the United Kingdom has been investigating the acute effect of two berry extracts on cognition and mood in adults aged 40 to 60 (NCT02810769). In this study, the participants are supplemented with two different berry-based formulations containing 500 mg of polyphenols, and after administration, their behavioral effects are measured through multiple cognitive assessments throughout the day. The study aims to assess whether acute supplementation with two berry extracts can enhance memory, attention, and executive function in an older population [324]. A randomized, placebo-controlled, double-blind clinical trial conducted in a collaboration between France and Canada has explored the cognitive effects of a polyphenol-rich dietary supplement in healthy adults (NCT02063646). The study aims to evaluate the impact of daily supplementation with the polyphenol-rich product over a 6-month period, compared to a placebo, on cognitive function [325]. The effects of the daily consumption of blueberry polyphenols on vascular function and cognitive performance (BluFlow) have also been investigated (NCT04084457). This randomized, double-blind, placebo-controlled, parallel trial conducted in the United Kingdom, involving healthy men and women aged 65 to 80 years, supports the hypothesis that the daily consumption of an anthocyanin-rich blueberry beverage can improve cognitive performance and vascular function by increasing cerebral blood flow in healthy elderly individuals [326]. Another investigation conducted in the United Kingdom seeks to elucidate the mechanisms involved in the acute and chronic cognitive effects of flavanol/anthocyanin intervention in humans (NCT03030053). This double-blind, randomized, controlled, chronic intervention parallel-arm trial, conducted with healthy elderly individuals, is being carried out to determine the effect of a flavonoid-rich supplement on cognitive function, peripheral arterial health, and brain mechanisms [327]. Furthermore, another study in the United Kingdom is exploring the impact of consuming cocoa products rich in polyphenols over 8 weeks on the cognitive function of individuals aged 50 and older (NCT02996578). The goal is to assess whether cocoa polyphenols enhance cognitive function and if these improvements are related to the changes in risk factors for cognitive decline associated with aging [328].

A clinical investigation conducted in Sweden is also evaluating the effects of daily consumption of berries and vegetables over 5 weeks on the cardiometabolic risk markers and cognitive functions in healthy elderly volunteers (NCT01562392) [329]. In another clinical investigation conducted in the United States, the supplementation of Aronia Berry to improve vascular endothelial dysfunction and modulate gut microbiota in middle-aged/elderly adults is being explored (NCT03824041) [330]. Another ongoing study conducted in Australia is evaluating the long-term effects of two plant-based, polyphenol-rich dietary supplements on cardiovascular health and low-grade inflammation in the elderly (NCT04763291) [331]. A new study exploring the benefits of blueberry consumption on improving vascular endothelial function in postmenopausal women with high blood pressure is underway. Conducted by researchers in the United States, the investigation evaluates the effects of a 12-week treatment period (NCT03370991) [332]. Moreover, a new ongoing trial also in the United States is investigating the antihypertensive and vascular protection effects of wild blueberries in middle-aged/elderly men and postmenopausal women (NCT04530916) [333].

Another study, also conducted in the United States, has been dedicated to investigating the impacts of a phytochemical supplement on “metabolic aging” in older overweight adults compared to young lean adults (NCT04919876) [334]. An interesting ongoing study in Canada seeks to evaluate the impact of Urolithin A (Mitopure) on mitochondrial quality in the muscle of frail elderly individuals (NCT06556706) [335]. Although Urolithin A is a product of bacterial fermentation, some polyphenol-rich foods with ellagic acid, including strawberries (Fragaria annassa), Java plums (Eugenia jambolana), and pomegranate fruits (Punica granatum), have the potential to stimulate the production of this active compound through colonic fermentation, with promising effects on metabolic health [336,337,338,339,340]. Thus, studies that elucidate its potential in relation to aging are of great relevance for developing strategies that could organically stimulate the production of this substance through the consumption of polyphenol-rich foods containing ellagic acid.

A clinical trial in Norway is investigating the polyphenol-based product “DailyColors™” (NCT05829382) in adults aged 55–80. The study aims to assess the effects of acute treatment on healthy aging by analyzing key blood biomarkers [341]. A pilot study conducted in the U.S. with 50 participants (NCT05234203) is investigating the impact of a polyphenol-rich nutritional supplement on the epigenetic and cellular markers of immunological age. The goal is to assess how the supplement influences epigenetic immune age and immune cell patterns over a 90-day period [342]. A pilot study currently underway in the United Kingdom is assessing the effects of taxifolin/dihydroquercetin and ergothioneine present in the diet on immunological biomarkers in healthy volunteers over 50 years old (NCT05190432) [343]. An ongoing parallel randomized trial also conducted in the United Kingdom is evaluating the effects of pomegranate extract on inflammaging (NCT05588479) [344].

Another investigation conducted in Spain is examining the effects of Pomanox^®^ supplementation, a standardized extract obtained from pomegranate fruit (*Punica granatum* L.), on skin aging (NCT05842447). The primary objective of the study is to assess the impact of consuming two doses of Pomanox^®^P30 on hyperpigmented skin spots in humans [345]. A randomized, double-blind, placebo-controlled trial in Spain has been investigating the effects of Oligopin^®^, a commercial extract from French maritime pine bark, on skin aging (NCT04141059). The primary aim of the study is to assess the potential benefits of Oligopin^®^ in improving skin elasticity in the participants with photo-aged skin [346].

Finally, a new ongoing study seeks to propose a new strategy to better understand interindividual variability in response to the diets of the elderly, based on deep phenotyping and thus establish a new multidimensional phenotyping for personalized preventive nutritional support for the elderly (NCT06163794) [347].

Overall, these findings reinforce the effectiveness of polyphenols in modulating the numerous imbalances associated with aging. However, more research is needed to confirm their effectiveness in human trials and explore their full therapeutic potential for age-related conditions.

**Table 2 nutrients-16-03621-t002:** Clinical evidence regarding the benefits of polyphenols in aging.

Foods/Polyphenols	Eligibility Criteria	Intervention/Treatment	Time Intervention	Results/Objectives	NCT Number	References
Cocoa polyphenols	Males/Females (50–60 years old)	• Polyphenol rich chocolate bar (581.4 mg)	8 weeks	Positive effects on facial wrinkles and elasticity, suggesting that cocoa flavanol supplementation may contribute to preventing the progression of photo-aging	NCT02060097	[318]
		• Polyphenol rich cocoa powder (554 mg)				
		• Low polyphenol content chocolate bar (198.5 mg)				
		• Low polyphenol content cocoa powder (191.2 mg)				
Red beet juice	Males/Females (40–65 years old)	• Red beetroot juice	4 weeks	Modulation of postprandial endothelial function and other cardiometabolic responses in overweight/obese individuals over 50 years old	NCT02949115	[319]
		• Red beetroot juice without nitrate				
		• Placebo drink plus potassium nitrate				
		• Placebo drink				
Cranberry	Males/Females (50–80 years old)	• Freeze-dried cranberry powder	12 weeks	Improves episodic memory performance and neural functioning	NCT03679533	[320]
		• Placebo				
Pomegranate juice	Males/Females (50–75 years old)	• Pomegranate juice	12 months	Stabilizes the ability to learn visual information	NCT02093130	[321]
		• Placebo				
Blueberries	Females (45–65 years old)	• Freeze-dried blueberry powder	8 weeks	Reduce blood pressure and arterial stiffness, effects that were partially attributed to increased nitric oxide production	NCT01686282	[322]
		• Placebo				
Curcuminoids	Males/Females (18–35 or >60 years old)	• New curcumin formulations		Bioavailability is markedly increased by micellar solubilization	NCT01982734	[323]
Berry extracts	Males/Females (40–60 years old)	• Berry drink	Change from baseline 60, 150, 240 and 360 min post dose	To assess whether acute supplementation with two berry extracts can enhance memory, attention, and executive function in an older population	NCT02810769	[324]
		• Control				
		• Powdered berry drink				
Polyphenol-rich dietary supplement	Males/Females (60–70 years old)	• Polyphenol-rich extract	6 months	To evaluate the impact of daily supplementation with the polyphenol-rich product over a 6-month period, compared to a placebo, on cognitive function	NCT02063646	[325]
		• Placebo				
Anthocyanin-rich blueberry beverage	Males/Females (65–80 years old)	• Wild blueberry powder	12 weeks	To evaluate the impact of daily consumption of an anthocyanin-rich blueberry beverage on cognitive performance and vascular function by increasing cerebral blood flow in healthy elderly individuals	NCT04084457	[326]
		• Placebo				
Flavonoid-rich supplement	Males/Females (60–75 years old)	• Cocoa-flavanol supplements	24–36 weeks	To determine the effect of a flavonoid-rich supplement on cognitive function, peripheral arterial health, and brain mechanisms	NCT03030053	[327]
		• Control supplements				
Cocoa polyphenols	Males/Females (50–60 years old)	• Polyphenol rich chocolate bar (581.4 mg)	8 weeks	To assess whether cocoa polyphenols enhance cognitive function and if these improvements are related to changes in risk factors for cognitive decline associated with aging	NCT02996578	[328]
		• Polyphenol rich cocoa powder (554 mg)				
		• Low polyphenol content chocolate bar (198.5 mg)				
		• Low polyphenol content cocoa powder (191.2 mg)				
Daily consumption of berries and vegetables	Males/Females (50–70 years old)	• Berries and vegetables	5 weeks	To evaluate the effects of daily consumption of berries and vegetables over 5 weeks on cardiometabolic risk markers and cognitive functions in healthy elderly volunteers	NCT01562392	[329]
		• Control product				
Aronia Berry	Males/Females (45–75 years old)	• Aronia full spectrum— half dose	6 weeks	To explore the benefits of supplementation of Aronia Berry on vascular endothelial dysfunction and gut microbiota modulation in middle-aged/elderly adults	NCT03824041	[330]
		• Aronia full spectrums— full dose				
		• Placebo				
Polyphenol-rich dietary supplements	Males/Females (55–80 years old)	• Juice plus+ fruit, vegetable and berry blends	12 months	To determine the long-term effects of two plant-based, polyphenol-rich dietary supplements on cardiovascular health and low-grade inflammation in the elderly	NCT04763291	[331]
		• Juice plus+ omega blend				
Blueberry	Females (45–65 years old)	• Blueberry powder	12 weeks	To explore the benefits of blueberry consumption on improving vascular endothelial function in postmenopausal women with high blood pressure	NCT03370991	[332]
		• Placebo powder				
Blueberry	Males/Females (45–70 years old)	• Blueberry powder	12 weeks	To investigate the antihypertensive and vascular protection effects of wild blueberries in middle-aged/elderly men and postmenopausal women	NCT04530916	[333]
		• Placebo powder				
Phytochemical supplement	Males/Females (≥55 years old)	• Fruit/vegetable supplement	6 weeks	To evaluate the impacts of a phytochemical supplement on “metabolic aging” in older overweight adults compared to young lean adults	NCT04919876	[334]
		• Placebo				
Urolithin A (Mitopure)	Males/Females (65–85 years old)	• Mitopure (Urolithin A)	8 weeks	To evaluate the impact of Urolithin A (Mitopure) on mitochondrial quality in the muscle of frail elderly individuals	NCT06556706	[335]
		• Placebo				
Polyphenol-based product (DailyColors™)	Males/Females (55–80 years old)	• DailyColors™	3 weeks	To assess the effects of acute treatment on healthy aging by analyzing key blood biomarkers	NCT05829382	[341]
		• Placebo				
Polyphenol-rich nutritional supplement	Males/Females (18–85 years old)	• HTB rejuvenate	90 days	To assess how the supplement influences epigenetic immune age and immune cell patterns	NCT05234203	[342]
Taxifolin/dihydroquercetin and ergothioneine	Males/Females (50–65 years old)	• Taxifolin	8 weeks	To investigate the effects of taxifolin/dihydroquercetin and ergothioneine present in the diet on immunological biomarkers	NCT05190432	[343]
		• Ergothioneine				
		• Control				
Pomegranate extract	Males/Females (60–70 years old)	• Pomegranate extract	6–12 weeks	To evaluate the effects of pomegranate extract on inflammaging	NCT05588479	[344]
		• Control				
Pomanox^®^	Females (30–65 years old)	• Pomanox (367 mg)	12 weeks	To assess the impact of consuming two doses of Pomanox^®^ on hyperpigmented skin spots in humans	NCT05842447	[345]
		• Pomanox (700 mg)				
		• Control				
Oligopin^®^	Males/Females (≥35 years old)	• Oligopin	6 weeks	To assess the potential benefits of Oligopin^®^ in improving skin elasticity in participants with photo-aged skin	NCT04141059	[346]
		• Placebo				

## 8. Conclusions

In conclusion, the health and longevity of an organism hinge on a delicate and dynamic balance between the processes of damage and repair, a balance that becomes increasingly precarious with age. This shift from homeostasis to homeodynamics underscores the complex interplay of the factors influencing aging, including the progressive decline in physiological functions and the heightened risk of chronic diseases. The intricate relationship between body composition and aging, particularly the roles of adipose tissue and muscle, is central to understanding the health challenges faced by the elderly. As the body ages, the redistribution of adipose tissue and its interaction with skeletal muscle significantly contribute to the onset of sarcopenia and mobility impairments. Cellular senescence and the associated inflammatory phenotype further exacerbate these issues, driving metabolic dysfunction at both the local and systemic levels. These challenges highlight the need for effective interventions to modulate these processes. Polyphenols, with their natural presence in a variety of foods, offer a promising approach to mitigating these age-related changes. Their inclusion in a diet presents a functional strategy with the potential for long-term benefits, as evidenced by the growing body of scientific literature. The therapeutic potential of polyphenolic compounds in restoring balance within the tissues affected by aging is particularly compelling, suggesting a path forward in the development of strategies to enhance quality of life and reduce morbidity and mortality in the elderly.

## Figures and Tables

**Figure 1 nutrients-16-03621-f001:**
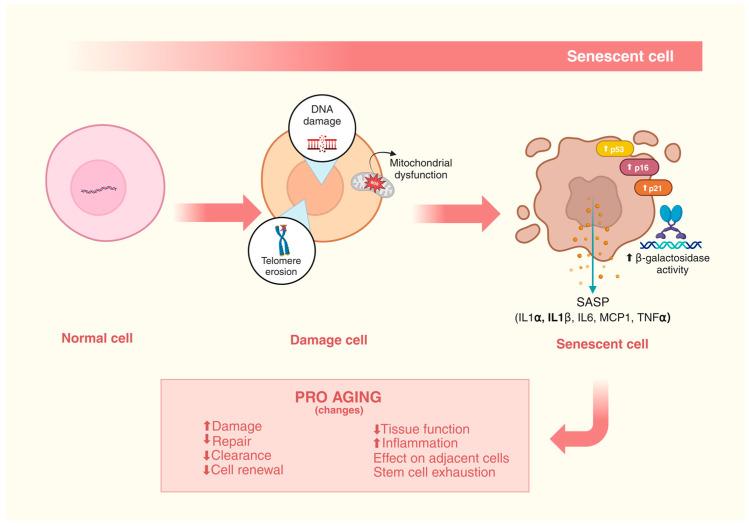
DNA damage, telomeric erosions, and mitochondrial dysfunction culminate in cellular damage that favors the cellular senescence process, mediated by increased beta-galactosidase activity, the expression of p53, p16, and p21, and the secretion of a SASP observed by the expression of IL1α, IL1β, MCP1, and TNFα. These cellular damages favor pro-aging changes, such as increased damage and inflammation, reduced repair, clearance, cell renewal, and tissue function, causing effects on adjacent tissues and the exhaustion of stem cells.

**Figure 2 nutrients-16-03621-f002:**
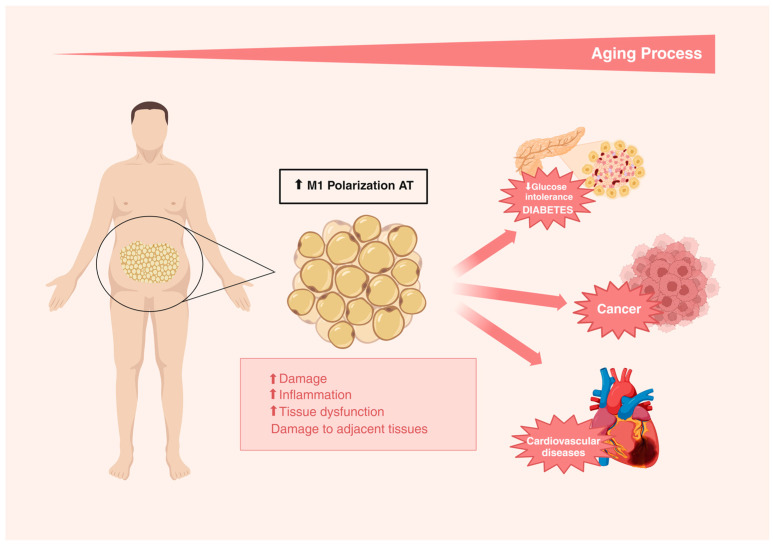
The increase in adipose tissue deposition related to the aging process is mediated by the increase in M1 polarization in adipose tissue, favoring increased damage, inflammation, tissue dysfunction, and damage to adjacent tissues, which may contribute to the development of the chronic diseases associated with age such as the following: reduced glucose tolerance, diabetes, cancer, and cardiovascular diseases.

**Figure 3 nutrients-16-03621-f003:**
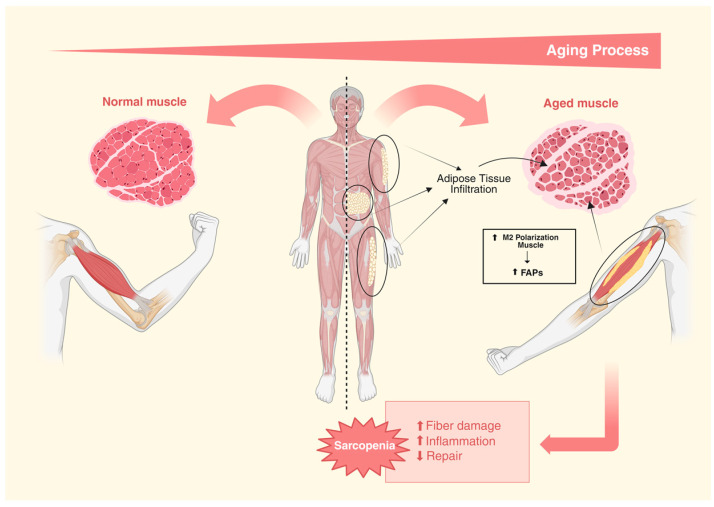
The infiltration of adipose tissue between muscle fibers during the aging process favors the increase in M2 polarization in muscle, leading to increased FAPs signaling, resulting in increased damage to muscle fibers, inflammation, and reduced tissue repair, corroborating the development of sarcopenia.

**Figure 4 nutrients-16-03621-f004:**
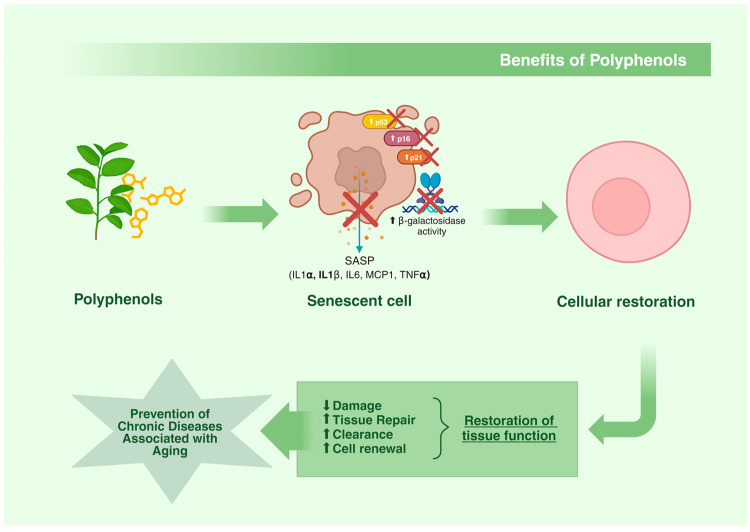
The use of polyphenols promotes benefits in cellular restoration, inhibiting senescence signaling, through damage reduction, increased tissue repair, clearance, and cellular renewal, culminating in tissue restoration and favoring the prevention of the chronic diseases associated with aging.

## Data Availability

Not applicable.

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
