# Peer review of "Body Composition and Senescence: Impact of Polyphenols on Aging-Associated Events"

_nutrients, 2024, doi:10.3390/nu16213621_

Round 1
Reviewer 1 Report
Comments and Suggestions for Authors
This great review discusses both (mainly cellular) senescence, and a possible treatment. The article is structured and comprehensive, along with an elaborate reference list. I have some remarks:
- An extra paragraph should be added, briefly discussing the Hallmarks of Aging (Lopez-Otin, C - Cell - 2013 and 2023).
- In paragraph 4, a sub-heading concerning cancer could be added.
- In paragraph 5, only flavonoids are discussed, although a sub-heading concerning tannins could be added.
Comments on the Quality of English LanguageSome typo's throughout the text.
Author Response
The revised version of the manuscript has successfully addressed all the points. Below are the specific points and how they were addressed in the updated manuscript:
- “An extra paragraph should be added, briefly discussing the Hallmarks of Aging (Lopez-Otin, C - Cell - 2013 and 2023)”.
As suggested, the authors have included a new paragraph that briefly discusses the Hallmarks of Aging, referencing both the 2013 and 2023 works by Lopez-Otin et al. This addition provides a broader context for understanding cellular senescence, and it has been appropriately highlighted in red in the current version.
- “In paragraph 4, a sub-heading concerning cancer could be added”.
The authors have added a sub-heading concerning cancer in paragraph 4, as recommended. This change improves the organization of the manuscript, allowing for a clearer discussion of the relationship between senescence and cancer. The addition has also been highlighted in red.
- “In paragraph 5, only flavonoids are discussed, although a sub-heading concerning tannins could be added”.
In response to the suggestion to broaden the discussion of compounds beyond flavonoids, the authors have added a sub-heading on tannins in paragraph 5. This enriches the treatment section by providing a more comprehensive overview of bioactive compounds, and the inclusion has been clearly marked in red.
Overall, the authors have implemented the suggested changes effectively, and the manuscript is now more structured and comprehensive. I appreciate the effort made to enhance clarity and depth, and the highlighted revisions in red make the improvements easy to track.
Reviewer 2 Report
Comments and Suggestions for Authors
This review was aimed to discuss how factors such as cellular senescence and the presence of a pro-inflammatory phenotype can negatively impact body composition and lead to the development of age-related diseases, as well as how the use of polyphenols can be a functional measure for restoring balance, maintaining tissue quality and composition, and promoting health. The reviewer considers the contents of this review to be useful clinical information from the perspective of preventive cellular senescence. Furthermore, the reviewer considers that this review thoroughly examines previous studies and summarizes it appropriately. However, the reviewer would like the authors to consider the following points.
The title of this review is "Polyphenols and Senescence: Impact of Body Composition on Aging-Associated Events."However, this review mainly discusses the effect of body composition on senescence. Given the content of this review, the reviewer believes that the title "Body Composition and Senescence: Impact of Polyphenols on Aging-Associated Events" would be appropriate. If the authors intend to keep the current title, the reviewer thinks that the main focus of the article should be on polyphenols.
The reviewer believes that readers would be further interested to provide the illustrations (figure) for mechanism of body composition on senescence, as well as the effects of polyphenol intake on aging-related events.
The reviewer considers that the results of clinical trials on polyphenol intake (P12, L592-P13, L678) would be better understood if the results of previous studies were summarized in a table.
Author Response
The revised manuscript has addressed all the points raised in my previous review, and I commend the authors for their diligent revisions. Below is a summary of the addressed points and how they have been incorporated into the updated version:
Title Revision:
The authors have responded to the suggestion regarding the title. The new title, "Body Composition and Senescence: Impact of Polyphenols on Aging-Associated Events", now accurately reflects the content of the review. This revision aligns the title with the article's main focus on body composition and polyphenols and has been highlighted in red.
Inclusion of Illustrations:
The authors have added illustrations that clarify the mechanisms of body composition's impact on senescence, as well as the effects of polyphenol intake on aging-related events. These visual aids enhance the comprehensibility of the manuscript, and the newly added figures are highlighted in red in the current version.
Summary Table for Clinical Trials:
As suggested, the authors have included a summary table for the results of clinical trials on polyphenol intake (P12, L592-P13, L678). This addition provides a clearer and more structured presentation of the data, making the results easier to interpret. The new table has been appropriately highlighted in red.
Overall, the revisions have significantly improved the clarity and organization of the manuscript. The authors have addressed all the points raised in the review, and the highlighted changes in red allow for easy identification of the updates. This version of the manuscript is now more informative and reader-friendly, and I recommend it for further consideration.